# Differentiable Robot Rendering

**Ruoshi Liu**[*1]   **Alper Canberk**[*1]   **Shuran Song**[2]   **Carl Vondrick**[1]

[1]Columbia University   [2]Stanford University

[drrobot.cs.columbia.edu](drrobot.cs.columbia.edu)

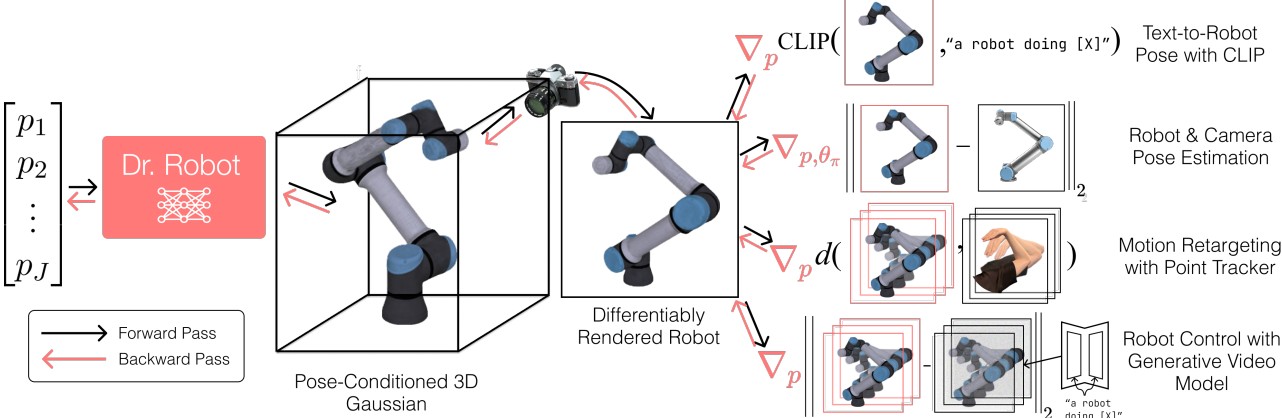

Figure 1: We introduce **D**ifferentiable **R**endering of **Robot**s (***Dr. Robot***), a robot self-model which is differentiable from its visual appearance to its control parameters. With it, we can perform control and planning of robot actions through image gradients provided by visual foundation models.

**Abstract:** Vision foundation models trained on massive amounts of visual data have shown unprecedented reasoning and planning skills in open-world settings. A key challenge in applying them to robotic tasks is the modality gap between visual data and action data. We introduce differentiable robot rendering, a method allowing the visual appearance of a robot body to be directly differentiable with respect to its control parameters. Our model integrates a kinematics-aware deformable model and Gaussians Splatting and is compatible with any robot form factors and degrees of freedom. We demonstrate its capability and usage in applications including reconstruction of robot poses from images and controlling robots through vision language models. Quantitative and qualitative results show that our differentiable rendering model provides effective gradients for robotic control directly from pixels, setting the foundation for the future applications of vision foundation models in robotics. Interactive demos and additional visualizations are available at: drrobot.cs.columbia.edu/.

**Keywords:** Robot Representation, Visual Foundation Model

## 1 Introduction

How to represent a robot? Classically, the appearance of a robot is modeled by a set of geometric primitives (e.g. triangle meshes), and the deformation of its morphology is modeled by its kinematic structure. In this paper, we introduce ***Dr. Robot***, a representation of robot self-embodiment based on Gaussians Splatting that is fully differentiable from its visual appearance to its control parameters.

---

[*] Equal contribution. Order can be listed either way.

8th Conference on Robot Learning (CoRL 2024), Munich, Germany.

The differentiability of a robot's self-model [1, 2, 3] is crucial. A differentiable representation allows us to pass the control signal from pixel space to control space through back-propagation. Having a differentiable self-model of the robot allows us to optimize for visual reward and constraints with gradients instead of resorting to gradient-free optimization methods such as evolutionary algorithms or reinforcement learning. This is particularly relevant today, given the growing evidence [4, 5, 6] that visual foundation models can provide robust and generalizable signals for control.

To obtain high-quality gradients, Dr. Robot tightly integrates three key components:

- **Gaussians Splatting.** The appearance of a robot in its canonical pose can be modeled by Gaussians Splatting, which models the robot's geometry and texture. Given a viewpoint, an image can be rendered differently through a differentiable rasterizer.

- **Implicit Linear Blend Skinning.** We adapt the traditional Linear Blend Skinning (LBS) technique in graphics to work with Gaussian splitting. Combining with differentiable forward kinematics, the implicit LBS model can project the positions of 3D Gaussians to target positions conditioned on a target pose.

- **Pose-Conditioned Appearance Deformation.** To model the change of appearance of a robot under various poses, we use a pose-conditioned appearance deformation model to change the spherical harmonics, scale, opacity, covariance matrix of the 3D Gaussians conditioned on a pose.

We validate the effectiveness of the gradients produced by our model by performing robot pose reconstruction from in-the-wild videos. Our optimization-based method outperformed the previous state-of-the-art method by a large margin. We also showcase various potential applications of Dr. Robot in several tasks, as shown in Fig. 1. By connecting Dr. Robot to various visual foundation models, we can perform planning and control of robot actions through optimization.

The primary contribution of this paper is a differentiable robot rendering method for vision-based robot learning. Our experiments demonstrate that our method provides a tool for solving complex robotic tasks through visual foundation models. We believe that as visual models, such as text-to-image and text-to-video models, continue to grow in spatial and physical reasoning ability, Dr. Robot will serve as the bridge between these foundation models and robotic control.

## 2 Approach

### 2.1 Overview and Formulation

Given a robot pose $\mathbf{p}$, we aim to model the visual appearance of the robot $\mathbf{I}$ from an arbitrary camera perspective as a differentiable function $f$,

$$\mathbf{I} = \pi(f(\mathbf{p})) \tag{1}$$

where $\pi$ denotes the image formation process under a pinhole camera model, which is a standard differentiable process. In the rest of this section, we will discuss how we formulate $f$ as a differentiable function. A differentiable rendering model for robots needs to have three key properties: full-body differentiability, deformability, and efficiency in rendering. We propose to model these three properties with forward kinematics (FK), linear blend skinning (LBS), and Gaussians Splatting (GS), respectively, as depicted in Fig. 2.

**Forward Kinematics.** Given a pose $\mathbf{p}$, we represent the position and orientation of joint $j$ with respect to the base joint 0 as the homogeneous matrix $T_j(\boldsymbol{p})$, which is the product of rigid transformations from parent joints along the kinematic chain:

$$T_j(\boldsymbol{p}) = \prod_{i=0}^{j-1} A_i(\boldsymbol{p}_i) \quad \text{for} \quad A_i(\boldsymbol{p}_i) = \begin{bmatrix} R_i^{i-1} & o_i^{i-1} \\ \mathbf{0} & 1 \end{bmatrix} \tag{2}$$

where $A_i$ denotes the coordinate frame of joint $i$ with respect to joint $i-1$.

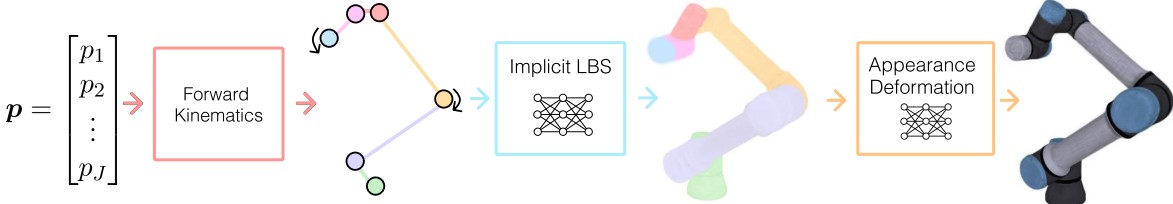

Figure 2: **Rendering Pipeline.** Our robot model is composed of 3 differentiable components. *Forward kinematics* projects a pose vector into a skeleton, *Implicit LBS* projects 3D Gaussians to the robot surface, and *Appearance Deformation* adjusts appearance of 3D Gaussians.

**Canonical Gausssians Splatting.** A robot in its canonical pose can be treated as a static 3D scene, which can be modeled by Gaussians Splatting (GS) [7]. Following the notation convention of the original paper [7], the canonical pose robot can be represented as a set of 3D Gaussians $\mathcal{G}$ with a set of means at $\{\mu_k\}$ where each Gaussian can be expressed as:

$$g_k = \mathcal{N}(\mu_k, RSS^T R^T) \tag{3}$$

where $RSS^T R^T$ denotes the full covariance matrix, R denotes the rotation matrix, and S denotes the scaling matrix. In addition, a 27-dimensional spherical harmonics coefficients $\mathbf{c}_{shs}$ and opacity $o$ are used to represent the apperance of the Gaussian during rendering. During optimization, after every N number of optimization steps, each Gaussian can split into two to densify the volume, and Gaussians with low opacity are truncated to reduce memory consumption.

**Implicit Linear Blend Skinning (LBS).** Canonical 3D Gaussian splatting can model the robot in a static pose, but we would like to render the robot at an arbitrary pose $\boldsymbol{p}$. Since optimizing a separate canonical 3D Gaussian for every pose is unrealistic, we need to construct a *geometric deformation* function to relocate the Gaussians in 3D space given a pose. Following SMPL [8], we learn a linear blend skinning (LBS) for modeling geometric deformations. Classical LBS expresses the transformation of each vertex on a mesh as a linear combination of joint transformations.

Different from [8], the splitting and truncation mechanism of 3D Gaussians during optimization makes it incompatible with the classical LBS, which assumes the input to be a fixed set of vertices. We therefore propose *implicit* LBS, which accepts an arbitrary 3D coordinate, and outputs a set of weights representing the influence of each of the $J$ joint transforms. Specifically, let $W(\mu) : \mathbb{R}^3 \to \mathbb{R}^J$ be the implicit LBS function, and $\mu$ be the position of one of the 3D Gaussians in canonical GS, the geometric deformation function $D$ can be expressed as:

$$D(\mu, \boldsymbol{p}) = \sum_{j=1}^{J} W(\mu)_j T_j(\boldsymbol{p})\mu \tag{4}$$

**Appearance Deformation.** So far, we've constructed an implicit LBS function to relocate 3D Gaussians according to robot poses, but a 3D Gaussian also contains appearance and shape information in addition to its 3D position, namely the rotation matrix $R_k$, scaling matrix $S_k$, and spherical harmonics coefficients $\mathbf{c}_{shs,k}$, and opacity $o_k$. Therefore, we additionally learn another appearance deformation function $X$ to predict the change of these parameters conditioned on the canonical and projected positions of 3D Gaussians:

$$(\Delta R_k, \Delta S_k, \Delta o_k, \mathbf{c}_{shs,k}) = X(\mu_k, \boldsymbol{p}) \tag{5}$$

Similar to the implicit LBS function, the appearance deformation function is also implicit, allowing them to share the same MLP architecture except for the input and output dimensions.

**Optimization.** Putting everything together, given the canonical GS $\mathcal{G}$ and a robot pose $\boldsymbol{p}$, the final posed GS $\mathcal{G}^{\boldsymbol{p}}$ can be expressed as:

$$f(\boldsymbol{p}) = \mathcal{G}^{\boldsymbol{p}} = X(D(\mathcal{G}, \boldsymbol{p}), \boldsymbol{p}) \tag{6}$$

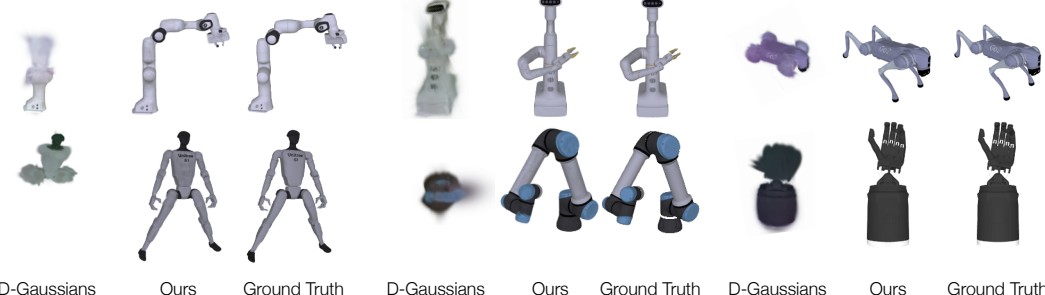

| D-Gaussians | Ours | Ground Truth | D-Gaussians | Ours | Ground Truth | D-Gaussians | Ours | Ground Truth |

Figure 3: **Visual Quality of Robot Model.** Here, we showcase the learned robot model's visual quality by comparing it with results obtained from Deformable Gaussians [9]. Due to the complicated kinematic structure of a robot, [9] is unable to fit the deformation while ours can.

Given a camera viewpoint $\pi$ and its corresponding visual observation $I$, we minimize the mean-squared error (MSE) between the prediction and observation to learn the full robot model:

$$\min_{\theta_X, \theta_W, \theta_G} L = ||I - \pi(X(D(\mathcal{G}, \boldsymbol{p}), \boldsymbol{p}))||_2 \tag{7}$$

**Test-Time Optimization** At test time, we can perform planning by optimizing robot actions through image gradients for various objectives, such as visual rewards calculated from visual foundation models. Please refer to Eq. 8, 9, 10 for examples of these optimization objectives.

## 3 Experiments

### 3.1 Self Model Quality

A robot self model needs to represent the appearance and morphology of the robot accurately. Therefore, in this section, we evaluate the faithfulness of the renderings and geometry when compared with ground truth against several baselines.

**Training data.** To train our model, we first curate a training dataset using Mujoco based on robot's URDF file composed of images of a robot under 10k poses, each captured from 12 viewpoints and split them into training and testing sets. We then use this data to train Dr. Robot by optimizing the objective function in Eq. 7.

**Evaluation Metrics.** Once training is finished, we evaluate the quality of the model on the testing set. We evaluate the appearance with PSNR score between the rendered images and ground truth. For geometry, we use the positions of the target pose Gaussians and evaluate the Chamfer distance between the ground truth pointcloud.

**Baselines.** We benchmark our results on 4 different baselines. Firstly, we compare against *Deformable Gaussians* [9], a state-of-the-art method to model deformable objects using GS. Secondly, we include *nearest neighbor*, where we retrieve the most similar rendering / pointclouds from 1000 samples for evaluation. Thirdly, we include *random*, where we draw a random sample from the training dataset for evaluation. *Nearest neighbor* and *random* baselines benchmark the difficulty of our tasks. Finally, we also include an ablated version of our model where the appearance deformation model is disabled as an ablation studies which we refer to as *no deform*.

**Results.** The quantitative results are shown in Table 1, and we also show examples in 3. Foremost, we observe that using LBS as a prior for geometric deformation has a significant positive impact on fidelity when compared with *Deformable Gaussians*, which uses K-plane [10] for modeling deformation. This gap is particularly notable for long robot arms with many chained joints such as the Panda Arm, where efficiently modeling movement in such high degrees through a neural network becomes infeasible.

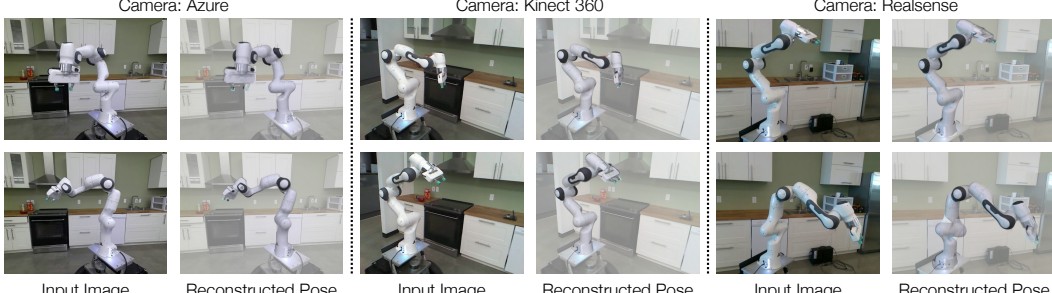

Figure 4: **Robot Pose Estimation from a Single Image.** From an input image, we perform optimization to reconstruct the joint angles of the robot and overlay the final rendering of the robot on top of the input image. These results show that accurate robot poses can be reconstructed from only a single image through our robot model. This also demonstrated that our robot model provides high-quality gradients for action optimization.

Table 1: **Visual and Geometric Quality of Rendering.** The table shows PSNR (higher is better) and Chamfer distance (lower is better) metrics for different methods and systems.

| PSNR (↑) | Shadow Hand | Unitree G1 | Unitree H1 | Franka Panda | Google Robot | ViperX 300 | xArm7 | Unitree GO1 | Unitree Go2 | UR5 |
|---|---|---|---|---|---|---|---|---|---|---|
| Ours | **31.44** | **28.31** | **27.77** | **32.84** | **32.60** | **30.59** | **33.25** | **29.74** | **29.60** | **32.01** |
| No Deform | 29.75 | 26.95 | 26.08 | 27.09 | 30.89 | 29.73 | 31.68 | 26.95 | 28.16 | 26.35 |
| K-plane | 9.74 | 11.23 | 8.26 | 16.52 | 13.21 | 8.45 | 18.93 | 11.23 | 8.41 | 13.63 |
| Nearest Neighbor (prev) | 13.66 | 13.07 | 10.19 | 20.68 | 18.32 | 14.13 | 25.10 | 10.80 | 14.87 | 15.44 |
| Nearest Neighbor | 14.81 | 15.58 | 12.20 | 22.40 | 19.64 | 16.38 | 27.12 | 11.66 | 15.84 | 17.61 |
| Random | 7.63 | 10.06 | 7.33 | 16.14 | 14.33 | 9.06 | 18.86 | 6.52 | 11.01 | 10.67 |
| **Chamfer Distance (↓)** | **Shadow Hand** | **Unitree G1** | **Unitree H1** | **Franka Panda** | **Google Robot** | **ViperX 300** | **xArm7** | **Unitree GO1** | **Unitree Go2** | **UR5** |
| Ours | **0.052** | **0.215** | **0.241** | **0.225** | **0.399** | **0.067** | **0.174** | **0.079** | **0.084** | **0.207** |
| No Deform | 0.054 | 0.251 | 0.263 | 0.315 | 0.434 | 0.074 | 0.190 | 0.085 | 0.089 | 0.104 |
| K-plane | 9.190 | 95.880 | 82.210 | 118.423 | 341.694 | 60.260 | 0.215 | 95.880 | 15.730 | 13.335 |
| Nearest Neighbor (prev) | 0.076 | 14.242 | 17.128 | 4.621 | 2.648 | 2.081 | 4.475 | 2.438 | 2.502 | 7.028 |
| Nearest Neighbor | 0.060 | 7.134 | 9.392 | 1.518 | 1.034 | 1.920 | 0.667 | 1.584 | 1.677 | 1.154 |
| Random | 0.429 | 45.580 | 50.872 | 51.325 | 17.342 | 28.374 | 79.229 | 3.611 | 3.591 | 111.021 |

## 3.2 Robot Pose Reconstruction

We evaluate the quality of gradients produced by Dr. Robot on a practical task – robot joint angles estimation from a single in-the-wild image. Given a monocular RGB image $I$ of a robot, we optimize jointly the robot pose $p$ and camera pose $\theta_\pi$ to minimize the following objective function,

$$\min_{p,\theta_\pi} \mathcal{L}_{\text{Rec}}(p, I) = ||I - \pi(f(p))||_2 \tag{8}$$

**Evaluation Protocols.** Following prior works [11, 12], we used the Panda-3CAM dataset composing around 10K images with corresponding robot poses created by [11] for evaluation. We evaluate the reconstruction results by calculating the L1 distance between the estimated joint angles and the ground truth joint angles. The dataset is composed of three long videos; we, therefore, perform optimization from scratch on the first frame of the video and warm start later frames with joint and camera poses from the previous frame as initialization, similar to what's done in both [11, 12].

**Baseline.** We compare against the state-of-the-art method on this task – *Robopose* [12]. Robopose estimates robot poses from a single image by a refiner neural network which takes in the image and the current rendering of the robot from the CAD model of the robot and output the delta joint angles.

**Results.** Quantitatively, we outperforms the previous SOTA method [12] by 32.9%. We perform consistently better across three camera platforms with different fields of view and mounting positions. As shown in Fig. 4, our reconstruction is highly accurate and is able to event reconstruct the gripper poses when they are visible which is challenging for prior works.

While camera and robot pose are both unknown in the preceding experiments, our method allows for reconstruction with known camera or robot parameters, or even using multiple views at once.

Table 2: Single-View Robot Pose Reconstruction

| Dataset | Ours | Ours (std) | Robopose | Robopose (std) |
|---|---|---|---|---|
| Panda-3cam Azure | **9.33** | 2.65 | 16.8 | 3.38 |
| Panda-3cam Kinect360 | **12.52** | 3.58 | 14.37 | 3.69 |
| Panda-3cam Realsense | **17.83** | 5.5 | 24.57 | 3.71 |

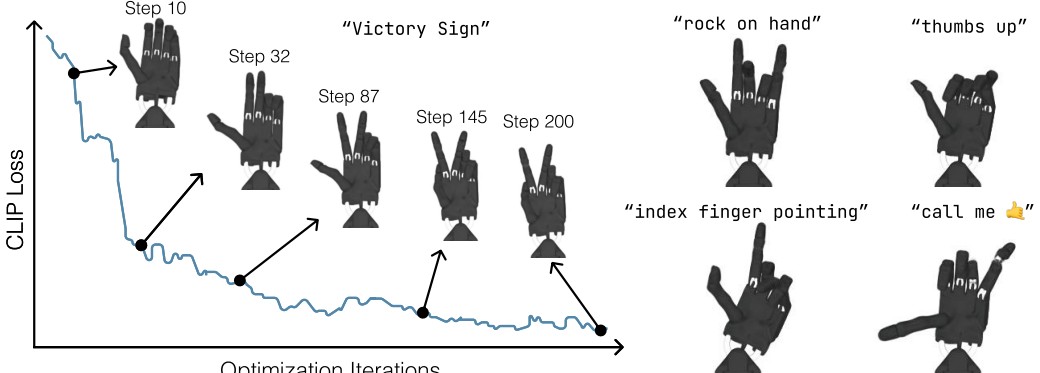

Figure 5: **Text-to-Robot Hand Gestures** We perform optimization of joint angles of a Shadow Hand to maximize the CLIP similarity between the rendered image and text prompt. We show the optimization process (**left**) as well as final outputs of different prompts (**right**).

## 3.3 Applications

In this section, we connect Dr. Robot with vision foundation models through image gradients. Different from Sec. 3.2 where we reconstruct robot poses from images/videos, here we show how visual foundation models can help robot *plan future actions* by passing gradients from the robot's visual appearance to its control parameters.

### 3.3.1 Text to Robot Pose with CLIP

Contrastive language-image models pretrained on internet data have shown remarkable knowledge of open-world images. In this demonstration, we leverage this ability to optimize the similarity of a robot hand to what CLIP judges to be similar to the given text prompt. In particular, let CLIP be denoted by $\mathcal{C}$, which takes in an image and a language embedding, returns the content similarity. Since CLIP is completely constructed from neural networks, it is fully differentiable. Furthermore, let $\pi$ be our differentiable rasterizer. We optimize the joint angles $\boldsymbol{p}$ for the following objective

$$\min_{p} \mathcal{L}_{\text{CLIP}}(\boldsymbol{p}; \text{text prompt}) = \mathcal{C}(\text{text prompt}, \pi(f(\boldsymbol{p}))) \tag{9}$$

As shown in Fig. 5, with only gradients provided by CLIP, we were able to perform optimization in the 24-dimensional action spaces of Shadow Hand to obtain semantically meaningful hand poses that correspond to given language prompts. These results demonstrate that Dr. Robot can be directly connected with off-the-shelf large-scale vision-language models (VLM) through image gradients.

### 3.3.2 Text to Action Sequences with Video Model

Recent works [13, 5] have shown that generative video models can be used for planning and guiding robot behavior. Since action data is not in the pre-training of these models, these approaches train their own inverse dynamics models to estimate robot poses in the generated videos. The ability of Dr. Robot to reconstruct images and videos in section 3.2 makes it an easy drag-and-drop replacement for an inverse dynamics model.

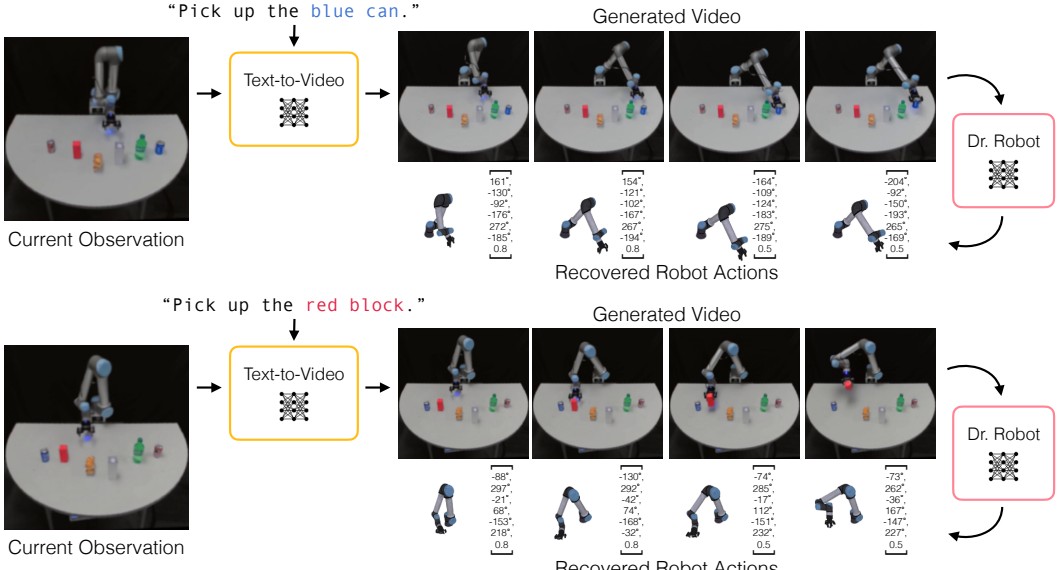

Figure 6: **Dr. Robot for text-conditioned robot control.** We show that robot actions can be extracted from videos predicted by a finetuned text-to-video model and be executed for robot control.

To demonstrate this application, we fine-tune Stable Video Diffusion on 100 episodes from the ASU Table Top dataset [14] conditioned on language embeddings of prompts following implementation of [15]. At test time, we are able to condition the video model on novel language prompts, and we reconstruct the robot by optimizing the reconstruction loss $\mathcal{L}_{\text{Rec}}$ outlined in section 3.2. We display samples from this process in figure 6.

### 3.3.3 Visual Motion Retargeting with Point Tracking

Motion retargetting [16, 17] is a long-standing problem in robotics that studies transferring motion from one embodiment to another. In prior works, this is typically done through estimating and matching the kinematic structure, e.g., between a humanoid robot and the human body. Here, we explore a new way to perform motion retargeting by matching the keypoint trajectories detected by video point trackers [18]. By using the point tracking model $\mathcal{P}$ and a differentiable renderer $\pi$, we can track the initial points $c_0^r$ on the robot across time to obtain $c_{1:T}^r = \mathcal{P}(c_0^r, \pi(\boldsymbol{p}_{1:T})) \in \mathbb{R}^{T \times C_2 \times 2}$. Our goal is to maximize the similarity between the tracked robot points $c_{1:T}^r \in \mathbb{R}^{T \times C_1 \times 2}$ and the goal points $c^{1:T} \in \mathbb{R}^{T \times C_2 \times 2}$. Since these points lack correspondence, we minimize the sum of Chamfer distances across each time step:

$$\mathcal{L}_{\text{Track}}(\boldsymbol{p}_{1:T}; c_0^r, c_{1:T}) = \sum_{1=t}^{T} d(c_t, c_t^r) = \sum_{1=t}^{T} d(c_t, \mathcal{P}(c_0^r, \pi(\boldsymbol{p}_{1:T}))_t)) \qquad (10)$$

As shown in figure 7, gradients produced by the sum of Chamfer distance between two sets of point tracks can be back-propagated through the point tracker, the image, and the robot model to optimize 24/37-dimensional control parameters. Since our method only requires test-time optimization, we relax the need for domain-specific training data, which is very hard for traditional motion re-targeting.

## 4 Related Works

**Differentiable Rendering** Rendering transforms 3D representations into 2D images, with differentiable rendering enabling gradient calculations via images. Research in this domain explores differentiable rendering using explicit 3D representations like meshes [19, 20], voxels [21, 22], and point

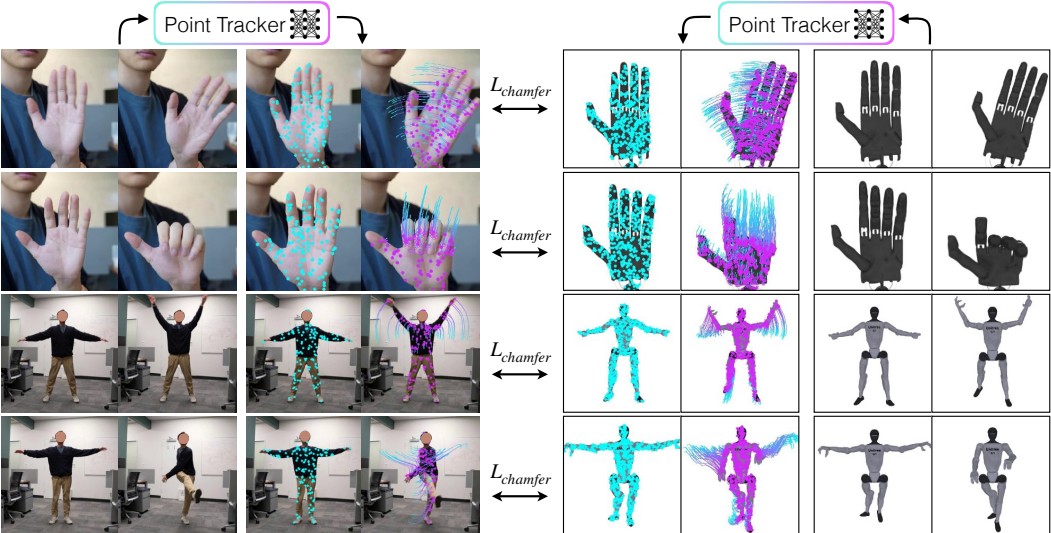

Figure 7: **Motion Retargeting with Video Point Tracker.** We perform optimization on robot action trajectories to minimize the chamfer distance between the point tracks in the rendered video and a demonstration video, allowing us to transfer motion across the embodiment gap.

clouds [23, 24], often converting these discrete structures into probabilistic distributions to facilitate gradient calculations. Since 2019, neural 3D representations have become prominent, providing fully differentiable frameworks. Key developments include NeRF [25] for volumetric rendering, DeepSDF [26] parameterizing signed distance functions, and Occupancy Networks [27] for volume occupancy modeling, applied in tasks like 3D reconstruction [28, 29, 30]. Recent advancements address the limitations of explicit representations and slow neural rendering, with Gaussian Splatting [7] introducing efficient differentiable rasterization using 3D Gaussians. This approach has been extended to model dynamics [9], physics [31, 32], and deformable objects [33].

**Visual Foundation Models for Robotics** Recent advancements have been made in pre-training perception models within visuomotor policies to develop robust visual representations, focusing on video prediction and contrastive learning to grasp dynamics and causality crucial for physical interactions. Techniques include predicting future events from current observations [34, 35, 36, 37, 38] and improving visual representations in robotics through contrastive learning and masked autoencoding [35, 39, 40, 41, 42, 43, 44]. Studies also focus on deriving a generalizable reward function from visual pretraining for reinforcement learning [4, 45, 46].

With the rapid advancements in text-to-video generative models [47, 48, 49, 50, 51], there is renewed interest in utilizing internet-scale video pretraining for robotics [5]. Some research employs video generative models as world simulators [52, 13, 47], predicting future scenarios conditioned on specific actions. Concurrently, video-language models are being used for longer-horizon planning [53, 54, 55]. In this paper, we propose an interface between these visual foundation models and the action parameters of robots, allowing planning and control of robot actions through image gradients.

## 5   Conclusion

In conclusion, our introduction of differentiable robot rendering bridges the modality gap between visual data and robotic action data, facilitating various applications of vision foundation models to robotic tasks. By integrating a kinematics-aware deformable model with Gaussian Splatting, our approach ensures compatibility across various robot form factors and degrees of freedom. Both quantitative and qualitative experiments confirm that our differentiable rendering model provides efficient and effective gradients for controlling robots directly from pixel data. We believe that Dr. Robot will become an effective tool for future applications with vision foundation models in robotics.

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

# A Implementation Details

## A.1 Model Architecture and Training Schedule

We initialize the canonical 3D Gaussians using 10,000 uniformly sampled point cloud on the robot mesh, and train the canonical Gaussians for 2,000 steps using the L1 image reconstruction loss. Simultaneously, we train our implicit LRS model using the chamfer distance between canonical and posed robot point clouds obtained from simulation. Finally, we train the canonical gaussians, the LRS model, and our appearance deformation network jointly on the L1 image reconstruction loss until convergence. The overall process takes 15-30 minutes on a single NVIDIA 3090 GPU depending on the robot embodiment.

Implicit LRS and appearance deformation modules of our method are modeled using 4-layer 256 hidden dimension lightweight MLPs. We encode coordinates using Fourier features as inputs to the networks.

We keep all canonical 3D Gaussian training hyperparameters consistent with the original 3D Gaussians codebase, and we use the same training hyperparameters across all robot embodiments.

## A.2 Text to Robot Pose with CLIP

In section 3.3.1 as our CLIP model, we use `openai/clip-vit-base-patch32` from Huggingface. We minimize the dot product between language and image embeddings that are output by their respective towers.

## A.3 Text to Action Sequences with Video Model

To obtain the generative video model shown in the paper, we fine-tune SVD-XT, which can generate 24-frame videos from an image. To enable language conditioning of this model, we replace the OpenCLIP `laion2b_s32b_b79k` model image embeddings with text embeddings from the same model. With this modification, we fine-tune this model on 256x256 videos from the OpenEmbodiment ASU dataset at 10fps following the implementation of [15] until convergence, which takes 12 hours on 2 A100 GPUs. The initial image conditioning and the prompt for the generated videos in the paper are chosen from a set of 12 validation episodes not seen during training.

