# OpenReview forum: "Differentiable Robot Rendering"
_robot-learning.org/CoRL/2024/Conference — CoRL 2024_

### Official Review · Reviewer_FkLZ · 2024-07-19
**Promising Concept with Significant Technical Limitations and Lack of Real-World Validation**

**Originality:** 4
**Technical Quality:** 3
**Clarity Of Presentation:** 3
**Potential Impact:** 3
**Recommendation:** 3
**Confidence:** 4

**Review:**

Strengths:

1. Novelty: The integration of Gaussian Splatting with robot kinematics is an innovative approach to creating differentiable robot models.
2. Potential Impact: If successful, this method could significantly advance the application of visual foundation models to robotics.
3. Versatility: The approach is designed to work with various robot form factors and degrees of freedom.
4. Performance: The reported 32.9% improvement over state-of-the-art in robot pose reconstruction is impressive.

Weaknesses:

1. Limited Real-World Validation: The paper lacks substantial testing on physical robots, which is crucial for demonstrating real-world applicability.
2. Simulation Dependence: Many experiments seem to rely heavily on simulated environments, raising questions about sim-to-real transfer.
3. Computational Complexity: The method relies on optimization at test time, which may be computationally expensive and slow for real-time robotic applications. The paper does not adequately address or evaluate computational efficiency.
4. Robustness: There's insufficient discussion on the robustness of the method to sensor noise, lighting variations, and other real-world factors.
5. Comparison: While the paper claims superiority over previous methods, the comparison is limited and not comprehensive enough, especially for text-to-robot control and motion retargeting applications.
6. Lack of Ablation Studies: The paper would benefit from thorough ablation studies to justify design choices and demonstrate the importance of each component.
7. Weak Theoretical Foundation: The paper lacks a rigorous theoretical analysis of the proposed method's properties, convergence, or optimality guarantees.
8. Sim-to-Real Gap: The paper does not address the potential challenges in transferring the learned models to real robots, which is a critical concern for practical applications.

The paper presents an intriguing concept with potentially significant implications for robot learning. However, it falls short in demonstrating its practical applicability in real-world robotics scenarios. The reliance on simulated environments and lack of extensive physical robot testing is a significant limitation.

The use of visual foundation models like CLIP for robot control is innovative, but the paper doesn't adequately address the potential limitations of such an approach, such as the possibility of hallucinated or unsafe actions, especially in more complex, multi-step tasks.
While the paper demonstrates improvements in robot pose reconstruction, it's unclear how this translates to improved robot performance in practical tasks. More concrete examples of how this approach enhances robot capabilities in real-world scenarios would strengthen the paper significantly.

**Quality Of The Limitations Section:**

2

**Questions For Rebuttal:**

1. Can you provide results from extensive testing on physical robots across various tasks and environments?
2. How does the computational complexity of your approach compare to existing methods, and is it feasible for real-time control on typical robotic hardware?
3. How robust is your method to real-world factors such as varying lighting conditions, occlusions, and sensor noise?
4. Can you provide a more comprehensive comparison with other state-of-the-art methods in robot control and pose estimation, especially for text-to-robot control and motion retargeting?
5. How well does your method generalize to more complex, multi-step robotic tasks beyond the simple scenarios demonstrated in the paper?
6. Can you provide ablation studies to justify your design choices and demonstrate the importance of each component in your framework?

**Robotics Focus:**

3

**Summary Of Paper:**

This paper introduces "Differentiable Robot Rendering" (Dr. Robot), a novel approach for creating a differentiable self-model of robots. It integrates Gaussian Splatting, Implicit Linear Blend Skinning, and Pose-Conditioned Appearance Deformation to enable full differentiability from visual appearance to control parameters. The authors demonstrate applications in text-to-robot pose generation, action sequence generation, and visual motion retargeting, aiming to bridge the gap between visual foundation models and robotic control.

**Summary Of Recommendation:**

While the paper presents an innovative approach with potential significance for robot learning, the lack of substantial real-world validation and several technical limitations are major concerns. The reliance on simulated environments, limited physical robot testing, and simplified scenarios make it difficult to assess the true impact and applicability of the method. The paper would be significantly strengthened by addressing these issues, providing more comprehensive real-world results, theoretical analysis, and ablation studies. In its current form, I recommend major revisions to address these concerns before consideration for acceptance.

---

### Official Review · Reviewer_trpR · 2024-07-20
**An interesting, novel tool for robot learning pipelines.**

**Originality:** 4
**Technical Quality:** 5
**Clarity Of Presentation:** 5
**Potential Impact:** 4
**Recommendation:** 4
**Confidence:** 4

**Review:**

This paper and video were very clearly presented. It is a paper of good quality given that is has great presentation and a novel task and method introduced.

Strengths:
The paper was easy and clear to read.
This is a very powerful tool for future robot learning researchers. The authors even demonstrate some simple ways to use the rendering pipeline for interesting robot performed tasks.
The rendered results look very similar to the ground truth robots.

Weaknesses:
It is not clear from the paper how novel the method is. The related works section does mention a number of related rendering paper, but it does not explicitly make comparisons with the presented work. I think that it would help the paper to move the related works section after the introduction.

I think that this work could be very significant in what it can enable for future methods for robot learning. The authors do a good job in demonstrating some simple tasks that can be performed using this renderer.

MISC:
Typo Line 139 "we outperforms" -> "we outperform"
Typo Line 217 "can connects" -> "can connect"

**Quality Of The Limitations Section:**

3

**Questions For Rebuttal:**

When optimizing the joint states, do you find that it is highly dependent on the initial joint states? For example, if the initial poses for Figure 4 are substantially far from the ground truth, will the optimization still find the best pose?

Could you please explain how your approach differs from any existing approaches? Is this perhaps the first approach?

**Robotics Focus:**

3

**Summary Of Paper:**

This paper presents a method for generating images of robots given their poses. The renderings are done in a differentiable manner. This enables some interesting downstream tasks to be possible such as optimizing robot pose to match a text prompt using a pre-trained image-text encoder.

**Summary Of Recommendation:**

This is a well written work that presents and interesting and useful new tool for robot learning.

---

### Official Review · Reviewer_VuJm · 2024-07-28
**Initial Review of Differentiable Robot Rendering Paper**

**Originality:** 4
**Technical Quality:** 4
**Clarity Of Presentation:** 4
**Potential Impact:** 3
**Recommendation:** 3
**Confidence:** 3

**Review:**

The paper is well-written and well-organized, with enough detail to understand the concepts and enough information in the supplementary material regarding the network architectures and the training procedure to reproduce the results. The authors combine several existing approaches (Gaussian Splatting, Linear Blend Skinning) but also extend them in a meaningful way to propose an end-to-end differentiable robot rendering model, that would be useful and relevant to the robotics community.

The experiments are sufficient and the quantitative results show that the proposed approach outperforms baselines in rendering quality and robot pose estimation. In addition, the authors present three applications of their methodology and conniving qualitative results (figures and video) for text-to-pose optimization using vision language models, recovering robot poses/actions from generative video model, and motion retargeting from human videos of hand/body to similarly structured robotic platforms.

Strengths:

-	Sound methodology and convincing experimental results show that the proposed method significantly outperforms baselines
-	Usefulness of the model in different kinds of vision-based robotic applications

Weaknesses:

-	Limited novelty in terms of methodology components (but with novel extensions and novel applications)
-	Only qualitative results reported for the application-related experiments

**Quality Of The Limitations Section:**

3

**Questions For Rebuttal:**

- For the self-model quality experiments, you are using nearest neighbor as a baseline, and you only use 1 000 samples from the training set (if I understand correctly) to find the nearest neighbor. However, your model is trained on a much larger training split from the 10k point clouds / 120k renderings respectively. Wouldn’t it be a fairer comparison if you use the complete training split for the nearest neighbor baseline when you compare to your method, and not only 1 000 samples?
- In the appendix you refer to LRS (do you mean LBS)?
- There are typos in lines: 139, 141 and 217

**Robotics Focus:**

3

**Summary Of Paper:**

The paper presents an approach for end-to-end differentiable rendering of robot images conditioned on joint angles and based on robot kinematics information. The method consists of 3 components, namely:  -	Robot’s forward kinematics is used to get a skeleton of the robot in a 2D image from specific viewpoint -	A modified version of Linear Blend Skinning (LBS) in combination with Gaussian Splatting is used to fit the position of the Gaussians for a pose-dependent the robot mesh -	Appearance deformation component is used to adjust the visual and shape information components of the Gaussians for further improvement of the rendering The authors show that their approach outperforms baselines in experiments related to rendering quality and robot pose reconstruction, and show several potential applications when combining the approach with foundation models.

**Summary Of Recommendation:**

The paper shows a methodology that enables end-to-end optimization of robot control policies from images/video and several interesting potential applications. There are no direct experiments on real robots but the paper demonstrates its applicability on different robotic platforms and due to this as well as due to the scope of the paper I think that hardware experiments are not necessary. Due to the sound methodology, the convincing quantitative and qualitative results and the potential applications I recommend accepting the paper.

---

### Decision · Program_Chairs · 2024-09-04

**Decision:**

Accept

**Comment:**

The reviewers agree that the paper is well-written and presents an innovative approach combining Gaussian Splatting with robot kinematics. They highlight the potential impact on bridging visual foundation models with robotics and the method's versatility across robot types. Quantitative results show significant improvements in rendering quality and pose estimation. However, concerns are raised about limited real-world validation, heavy reliance on simulations, and potential computational complexity. Reviewers suggest more comprehensive comparisons with existing methods, ablation studies, and addressing the sim-to-real gap. While the paper demonstrates interesting applications, more rigorous evaluation of practical robotic scenarios and computational efficiency is recommended to strengthen the work.
Following the rebuttal, reviewers comments were addressed, and reviewer FkLZ Raised their scorer to a weak accept.